

# What historical landfast ice observations tell us about projected ice conditions in Arctic Archipelagoes and marginal seas under anthro - pogenic forcing

Frédéric Laliberté[1], Stephen. E. L. Howell[1], Jean-François Lemieux[2], Frédéric Dupont[3] and Ji Lei[3]

[1]Climate Research Division, Environment and Climate Change Canada, Toronto, Ontario, Canada

[2]Recherche en Prévision Numérique Environnementale, Environnement et Changement Climatique Canada, Dorval, Québec, Canada

[3]Service Météorologique Canadien, Environnement et Changement Climatique Canada, Dorval, Québec, Canada

*Correspondence to*: Frédéric Laliberté (laliberte.frederic@gmail.com)

**Abstract.** Arctic landfast ice extent and duration from observations, ice assimilations, ocean re-analyses and coupled models are examined. From observations and assimilations, it is shown that in areas where landfast ice conditions last more than 5 months the first-year ice grows typically to more than 2 m and is rarely less than 1 m. The observed spatial distribution of landfast ice closely matches assimilation products but less so for ocean re-analyses and coupled models. Although models generally struggle to represent the landfast ice necessary to emulate the observed sea ice dynamics in regions favourable to landfast ice conditions, some do exhibit both a realistic climatology and a realistic decline of landfast ice extent under an an - thropogenic forcing scenario. In these more realistic simulations, projections show that an extensive landfast ice cover should remain for at least 5 months of the year well until the end of the 21st century. This is in stark contrast with the simula - tions that have an unrealistic emulation of landfast ice conditions. In these simulations, slow and packed ice conditions shrink markedly over the same period. In all simulations and in areas with landast ice that last more than 5 months, the end-of-winter sea ice thickness remains between 1 m and 2 m well beyond the second half of the century. It is concluded that in the current generation of climate models, projections of winter sea ice conditions in the Canadian Arctic Archipelago and the Laptev Sea are overly sensitive to the representation of landfast ice conditions and that ongoing development in landfast ice parametrization will likely better constrain these projections.

## 1 Introduction

Sea ice that is immobile because it is attached to land is termed "landfast". In shallow coastal regions, large pressure ridges can get anchored at the sea floor. These grounded ridges might then act as anchor points to stabilize and maintain a landfast ice cover [Mahoney et al., 2007]. However, landfast ice is also present in some coastal regions that are too deep for pressure ridges to become grounded. In this case, the ice can stay in place due to the lateral propagation of internal ice stresses that originate where the ice is in contact with the shore. Sea ice typically becomes landfast if its keel extends all the way to the



sea floor or if ice stresses cannot overcome lateral friction at the coastline [Barry et al., 1979]. Most (but not all) landfast ice
melts or becomes mobile each summer. Multi-year landfast ice (also termed an    "ice-plug") is rare but it is known to occur
within the Nansen Sound and Sverdrup Channels regions within the Canadian Arctic Archipelago (CAA) [Serson, 1972;
1974]. These ice-plugs were once a prominent feature within the CAA from the 1960s (Nansen Sound) and 1970s (Sverdrup
Channel) up until they were both removed during the anomalously warm summer of 1998 and have since rarely re-formed
[Alt et al., 2006]. The disappearance of multi-year landfast ice is coincident with a decline in pan-Arctic landfast ice extent
of approximately 7% decade-1 from 1976 to 2007 [Yu et al., 2013]. Landfast ice has not only shrunk in extent but has also
thinned. While few long-term records of sea ice thickness exist, they all show a thinning of springtime landfast ice. The
largest declines are generally found in the Barents Sea at 11 cm decade-1 [Gerland et al., 2008]. Along the Russian coast and
in the CAA, the thinning has generally been less pronounced and is on average less than 5 cm decade-1 [Polyakov et al.,
2010 for Russia, Howell et al., 2016 for Canada].
Landfast ice is immobile and, therefore, its maximum ice thickness is primarily driven by thermodynamics from air
temperature and the timing and amount of snowfall during the growth period [Brown and Cote, 1992]. Because it isolates
thermodynamics from dynamics, landfast ice is a convenient bellwether of the effect of anthropogenic forcing on the Arctic
environment. This convenience has motivated several studies that investigated the sensitivity of landfast ice to anthropogenic
forcing in both one-dimensional thermodynamic models [Flato and Brown, 1996; Dumas et al., 2006] and CAA-focused re -
gional three-dimensional ice-ocean coupled models [e.g. Sou and Flato, 2009]. Since the Sou and Flato [2009] study, several
high resolution global ocean and sea ice models have become available, thus making it possible to study the coupled re       -
sponse of landfast ice to anthropogenic forcing. These models include the Community Earth System Model Large Ensemble
(CESM-LE), coupled climate models from the Coupled Model Intercomparison Project phase 5 (CMIP5) and from the
Ocean Reanalysis Intercomparison Project (ORA-IP). Howell et al., [2016] provide a preliminary investigation of the afore  -
mentioned climate models within the CAA over a 50+ year record from 1957-2014 and found that they provide a reasonable
climatology but trends were unrealistic compared to observations.
In this study, we provide a more comprehensive investigation into variability of landfast ice extent and thickness from the
current generation of climate models for the Arctic-wide domain and also evaluate their response to anthropogenic forcing.
As climate models do not output a dedicated landfast ice variable and as the ice velocity does not completely vanish when
landfast ice is simulated, we first develop an approach to characterize landfast ice. We then describe the historical evolution
of landfast ice extent and springtime landfast ice thickness as well as their future projections in models. Finally, we compare
the coupled model simulations with our own pan-Arctic ice-ocean simulations.



## 2 Data Description

### 2.1 Observations

One of the longest records of landfast ice thickness and duration comes from several coastal stations throughout Canada that date back to the late 1940s, depending on the location. The dataset is available online at the Canadian Ice Service (CIS) web site (http://www.ec.gc.ca/glaces-ice/, see Archive followed by Ice Thickness Data). The thickness measurements are usually performed weekly from freeze-up to breakup, as long as it is safe to walk on the ice. For these reasons, the landfast ice dura - tion at these stations, measured as the number of weeks with measurements, is always biased on the shorter side, possibly by a few weeks. From these station records, we selected the four sites in the CAA that had continuous records up to 2015: Alert, Eureka, Resolute and Cambridge Bay. From these weekly records available from 1960 to 2015, we extracted the landfast ice duration and springtime landfast ice thickness. A thorough analysis of these quantities as derived from these records was pre - sented initially by Brown and Cote [1992] from 1957-1989 and recently updated to 2014 by Howell et al. [2016].

For additional ice thickness information we use ice thickness surveys in landfast regions of the CAA carried out by means of airborne electromagnetic induction (AEM) sounding in 2011 and 2015 previously described in Haas and Howell [2015]. These surveys were averaged on a 25 km EASE 2.0 grid and are shown in Figure S1 of the supplementary online material. We also use weekly ice thicknesses retrieved from CryoSat-2 / SMOS in netCDF format for the years 2010-2016, obtained from data.scienceportal.de and remapped using a nearest-neighbour remapping to a 25 km EASE 2.0 grid. The resulting win - ter maximum sea ice thicknesses are shown in Figure S2 of the supplementary online material.

In order to spatially map landfast ice we use the National Ice Center (NIC) ice charts products from the NSIDC (dataset ID G02172) and ice charts from the Canadian Ice Service Digital Archive (CISDA). The NIC ice charts are available from 1972 to 2007 but we restrict the time period to 1980-2007 to be consistent with CISDA. Indeed, the CISDA provide ice informa - tion before 1980 but landfast ice was not explicitly classified. We refer readers to Tivy et al. [2011] (CISDA) and Yu et al. [2014] (NIC) for in-depth descriptions of ice chart data and their validity as a climate record. Following Galley et al. [2010], who also used the CIS ice chart data to map landfast ice, we consider grid cells with sea ice concentration of 10/10ths to be landfast. We defined pan-Arctic landfast extent using the NIC ice charts (given their larger spatial domain) as the regions that are covered by landfast ice for at least one month in the climatology. Both the NIC and CISDA ice charts were converted from shape files to a 0.25 ° latitude-longitude grid and then converted using a nearest-neighbor remapping to a 25 km Equal- Area Scalable Earth (EASE) 2.0 grid. We compute the number of months (equivalent to "percent of the year" in Galley et al.) during which each grid cell was landfast for each time period from September to August.

### 2.2 Models

Climate simulations and reanalyses do not provide a variable that explicitly characterizes landfast ice conditions. This makes it challenging to verify how it emulates landfast ice conditions as compared to observations. To circumvent this limitation,



we use daily sea ice thickness (hereafter, sit), sea ice concentration (hereafter, sic) and sea ice velocities (hereafter, usi and
vsi) to synthetically characterize landfast sea ice conditions using the following procedure:
1.  On the original model grid, we set the land mask to its nearest neighbor and remap using a nearest neighbor remapping
usi, vsi and sit to the sic native grid. Finally, we use a nearest neighbor remapping to put all variables on a EASE 2.0
grid.
2.  The sea ice speed (hereafter, speedsi) is computed from usi and vsi on this new grid.
3.  Daily speedsi, sit and sic are averaged to weekly means.
4.  A grid cell is identified as having "packed ice" if the remapped weekly-mean sic is larger than 85%.
5.  A grid cell is identified as having "slow ice" if the remapped weekly-mean speedsi is less than 1 cm s-1 (~1 km day-1).
6.  Slow, packed ice is used as a proxy for landfast ice.

At each grid cell we then compute the number of months in each year with slow, packed ice. Using slow, packed ice is repre -
sentative because we are interested in one specific aspect of landfast ice: the fact that its growth is primarily driven by ther  -
modynamics and not by the sea ice dynamics. This procedure is used with the Pan-Arctic Ice-Ocean Modeling and Assimila -
tion System (PIOMAS) [Zhang and Rothrock, 2003], a subset of the highest resolution models [see Table 3, Storto et al.,
2011; Forget et al. 2015; Haines et al., 2014, Zuo et al., 2015; Masina et al., 2015] from the ORA-IP [Balsameda et al., 2015;
Chevallier et al., 2016]. Finally, we use the CESM-LE and CMIP5 models to analyze climatological landfast ice extent and
thicknesses.  Some ORA-IP models (ORAP5.0, UR025.4) do not provide daily output. For these models, monthly data was
first interpolated to daily frequency and from then on the analysis was performed using the procedure described above. It
should be noted that sea ice velocities are not provided by all models and only for a few simulations, constraining the scope
of the intercomparison presented here (see available models in Table 1). The data for this study was retrieved from the ESGF
using the cdb_query tool (github.com/cdb_query). Finally, the 1980-2005 Historical experiment followed by the 2006-2015
Representative Concentration Pathway 8.5 (RCP85) experiment [Taylor et al. 2012] are used with daily sea ice velocities,
thickness and concentration.

The models listed above do not represent the grounding of pressure ridges. Hence, they are not expected to perform well in
regions where grounding is known to be an important mechanism for the formation and stabilization of a landfast ice cover.
Observations show that grounding is important in the Laptev Sea  [Haas et al., 2005, Selyuzhenok et al., 2017], in the Beau -
fort Sea [Mahoney et al., 2007] and in the Chukchi Sea [Mahoney et al., 2014]. Nevertheless, these models can simulate
landfast ice in some regions because their dynamic takes into account mechanical interactions. For most of these sea ice
models, ice interactions are represented by a viscous-plastic rheology with an elliptical yield curve [Hibler, 1979].

Recently, a basal stress parameterization representing the effect of grounding was developed [Lemieux et al. 2015]. This pa -
rameterization calculates, based on simulated ice conditions, the largest ridge(s) at each grid point. When these subgrid scale



ridge(s) are able to reach the sea floor, a basal (or seabed) stress term is added to the sea ice momentum equation. This
grounding scheme clearly improves the simulation of landfast ice in regions such as the Alaskan coast, the Laptev Sea and
the East Siberian Sea. However, in deeper regions such as the Kara Sea, Lemieux et al. 2015 pointed out that their model
systematically underestimates the area of landfast. As the grounding scheme is less active in these deeper regions, Lemieux
et al. 2016 modified the viscous-plastic rheology to promote ice arching.

Following the work of Lemieux et al. 2016, we conducted simulations that combined the grounding scheme and a modified
viscous-plastic rheology. We used the optimal parameters k1=8 and k2=15 Nm-3 for the grounding scheme [Lemieux et al.
2015]. As opposed to the standard elliptical yield curve, the ellipse aspect ratio is set to 1.5 (instead of 2) and a small amount
of isotropic tensile strength is used (kt=0.05).

For these simulations, we used the ocean model NEMO version 3.1 and the sea ice model CICE version 4.0 with code modi -
fications to include the grounding scheme and to add tensile strength [Lemieux et al. 2016]. Our 0.25 ° grid is a subset of the
global ORCA mesh. It covers the Arctic Ocean, the North Atlantic and the North Pacific. This ice-ocean prediction system,
that includes tides, was developed as part of the CONCEPTS (Canadian Operational Network of Coupled Environmental
PredicTion Systems) initiative. We refer to our 0.25 ° model setup and simulations as CREG025 (CONCEPTS-regional
0.25°).
Note that while adding the tides to our ice-ocean prediction systems, we found that unrealistic sea thicknesses developed in
late winter in tidally active regions (e.g. Foxe Basin). To mitigate this problem, the Hibler 1979 ice strength parameterization
is used as opposed to the default Rothrock 1975 formulation. The ice strength parameter P* was set to 27.5 kNm-2 for our
CREG025 simulation.
The sea ice model was initialized with sea ice thicknesses and concentrations from the GLORYS2V1 ocean reanalyses. The
ocean model was initialized by the World Ocean Atlas (WOA13) climatology and forced at open boundaries by GLO -
RYS2V3 (Ferry et al. 2010; Chevallier et al., 2017). A spin up from October 2001 to September 2004 was performed. Free
runs (no assimilation) are then restarted from the fields in September 2004 and conducted up to the end of 2010. The simula -
tion was forced by 33 km Environment Canada atmospheric reforecasts [Smith et al. 2014].
**3 Results**
**3.1 Landfast ice duration and thickness**
The CAA is almost entirely covered by landfast ice for up to 8-months of the year (i.e. November to July) [Canadian Ice Ser -
vice, 2011] and is therefore a useful region to begin evaluating model representation of landfast ice duration and thickness.
Figure 1 illustrates the relationship between landfast ice thickness and duration within the CAA for the observed datasets
(e.g. CryoSat-2, AEM and in situ) in addition to PIOMAS and CREG025. When combining these heterogeneous data



sources, a general picture of their representativeness of ice thickness over landfast ice duration emerges. Based on in situ ob -
servations landfast ice within the CAA lasts from 4 to ~9 months grows to ~2 m which is in agreement with previous studies
[e.g. Brown and Cote, 1992; Canadian Ice Service, 2011; Howell et al., 2016]. For PIOMAS, CREG025 and CryoSat-2 ice
thickness standard deviations are close to the variability observed at the in situ locations. However, very thick ice upwards of
~4 m is encountered at the 95th percentile in both the CryoSat-2 and the PIOMAS data when the landfast ice lasts for more
than 9 months. These very stable and thick landfast conditions are the result of large multi-year ice floes, thus increasing the
average ice thickness. It has long been known that MYI forms in situ within the CAA and very thick MYI from the Arctic
Ocean is also advected into the region [e.g. Melling, 2002] which is evident from the airborne EM measurements thickness
values [Haas and Howell, 2015]. This mix of ice-types makes it challenging for models to represent ice thickness within the
CAA but overall, they are in reasonable agreement with observations.

## 3.2. Geographical distribution and climatology

The spatial distribution of annual landfast duration from observations (CIS and NIC), PIOMAS and selected ocean re-analy -
sis models is shown in Figure 2. Both ice charts products (CIS and NIC) show a similar landfast ice extent and duration in
the CAA (Figure 2a-b). This landfast ice extent is also very similar in the two ice chart products over their regions of overlap
(Figure 2a-b, magenta curve). In PIOMAS, the duration of slow and packed ice conditions, compares relatively well to the
overall landfast extent and duration in the ice chart products (Figure 2c). There is however, too much of the slow and packed
ice in the Beaufort Sea but too little in the Laptev and Kara Seas. Most ocean re-analysis products have a suitable representa -
tion of slow, packed ice conditions in the CAA, the notable exception being CGLORS and UR025.4 (Figures 2e-g). In the
CGLORS case, the ice component appears to still be in spin-up at the beginning of the integration period because there is an
unphysical interannual variability in the first few years of the simulation and therefore results should not be expected to con -
form to observations (Figure 2d). In the UR025.4 case, winter ice is packed but is too mobile in the Parry Channel and the
M'Clintock (Figure 2h).
The spatial distribution of annual landfast ice duration in CMIP5 models with higher resolution is illustrated in Figure 3b-h.
These models exhibit a reasonable slow, packed ice extent and duration but it is mostly confined to the CAA (Fig. 3b-h). The
exception is the MRI-ESM1 (and applies to the other models from the MRI) that simulate slow, packed ice conditions year-
round across the Arctic (Figure 3e). This is likely due to its sea ice being modeled as a simple viscous fluid, without a sophis-
ticated rheology. Compared to the NIC analyses, all the CMIP5 models and reanalyses do not have enough months of land -
fast ice on the Russian coast. GFDL-ESM2G , CESM-LE and PIOMAS are the ones that provide the best landfast ice simu -
lation in the Laptev, Kara and East Siberian Seas (Figure 2c-h; Figure 3f,h). Another important feature of sea ice dynamics in
coupled models (ACCESS 1.0, CESM-LE, GFDL-ESM2G) seems to be the tendency of many of them to emulate year-
round or close to year-round landfast ice in the Parry Channel regions of the CAA (Figure 3c,f,h). This is peculiar since this
would mean that ice likely takes years to transit through the Parry Channel, allowing thermodynamic forcing to create very



thick ice in a region. Note that in the remaining models, the MIROC5 and MPI-ESM-MR both emulate too short of a land-
fast ice duration in the Parry Channel.

## 3.3. Trends in landfast ice duration

The largest observed negative trends in landfast ice duration of up to 1 month decade-1 is found primarily in the East
Siberian Sea but a general negative trend is located across the Arctic (Fig. 4a-b) as also reported by Yu et al. [2014]. In the
CAA, trends are larger in the NIC ice charts but both the CIS and NIC show relatively weak duration declines in the Parry
Channel and the M'Clintock. These relatively small trends are in stark contrast with the very large trends almost everywhere
in the CAA in the PIOMAS simulations. For CGLORS, the model whose sea ice is still in spinup, there is a large positive in-
crease in slow, packed ice duration (Figure 4d). Such increases are also seen in the Beaufort Sea in the GLORYS2V3 re-
analysis indicating that towards the end of the reanalysis the Beaufort Sea is covered by slow, packed ice for a few months
per year (Figure 4g). This is in complete disagreement with observations and mandates that extra care be taken when using
this product to analyze the sea dynamics in the Beaufort Sea. In summary, re-analysis products appear to have a particularly
difficult time reproducing the long-term stability of the slow, packed ice distribution, suggesting that targeted efforts to im-
prove this aspect of their sea dynamics are likely necessary.
CMIP5 models sea ice simulations (except the MRI models for the reason explained above), on the other hand, fare rela-
tively well at representing negative trends in landfast ice duration when compared to observations (Figure 5). Most models
tend to show an enhanced disappearance of slow, packed conditions along the Beaufort Sea edge of the CAA and declines
that are in general agreements with observation in the Parry Channel. One exception is the CESM-LE where some of year-
round slow, packed ice conditions are not declining over the 1980-2015 period (Figure 5f). The models with less slow,
packed ice than in observations, MIROC5 and MPI-ESM-MR, show relatively strong declines that, if they continued, would
indicate an almost complete disappearance of slow, packed ice by mid-21st century.

## 3.4. Regional evaluation of landfast ice extent and thickness

We now take a closer regional examination at landfast ice extent in the CAA, Northwest Passage (Parry Channel route) and
Laptev Seas. These regions are expected to experience increases in shipping activity from the mid to late-21st century ac-
cording to model simulations [Smith and Stephenson, 2013; Melia et al., 2016]. Instead of using an absolute measure of ex-
tent, we report extent as a fraction of the ocean surface within the bounds of the NIC 1 month duration landfast ice extent cli-
matology (magenta line in Figure 2b). This approach is necessary to appropriately compare observations to models that rep-
resent the islands and channels of the CAA differently.
Over the 1980-2015 time period, landfast ice extent has declined dramatically for durations longer than 5 months with a
marked decline in the extent of landfast ice with a 7 to 8 months duration within the Northwest Passage (Figure 6). What is





however striking is how the extent of landfast ice extent with duration of 5 months or less has been mostly constant over the
last 35 years (Figure 6). If sea ice-albedo feedback is an important player in recent sea ice decline [e.g. Perovich et al., 2007]
then it is not entirely surprising that during the polar night landfast ice conditions re-establish themselves year after year
even in the context of rapid Arctic warming. Finally, it is also worth noting that Figure 6a indicates that the small amounts of
multi-year landfast ice within the CAA have virtually disappeared in recent years (i.e. the 11 months line is at 0 since 2002)
consistent with Alt et al., [2009].
Landfast ice extent in the CAA and Northwest Passage is well represented in the PIOMAS data assimilation product as it
compares well with the CIS and NIC ice chart products although, the NIC product does exhibit stronger interannual variabil-
ity (Fig. 7a-b). In the Laptev Sea, PIOMAS clearly underestimates the area of landfast ice when compared to the NIC ice
charts (Figure 7c). This is likely due to the fact that PIOMAS does not represent the effect of grounding, an important mech-
anism for the formation and stability of the Laptev Sea landfast ice cover [Selyuzhenok et al., in press]. Despite this too
small area of landfast ice in the Laptev Sea, PIOMAS exhibits a decline of ~25% of the landfast extent over the last 35 years
which is consistent with the one from the NIC ice charts.
Comparing current (1980-2015) to projected (2070-2080) landfast ice extent from CMIP5 in these regions reveals consider-
able changes which are summarized in Table 1. The seven models with the lowest extent of1979-2015 CAA slow, packed ice
(ACCESS1.0, ACCESS1.3, BCC-CSM1.1(m), GFDL-CM3, MIROC5, MPI-ESM-LR, MPI-ESM-MR) lose most of it by
2070-2080 while the four models with a large extent of 1979-2015 CAA slow, packed ice (CESM-LE, GFDL-ESM2G,
GFDL-ESM2M, NorESM1-M) retain most of it by 2070-2080. As mentioned earlier, two models have unrealistic behavior
(MIR-ESM, MRI-CGCM3) because their sea ice model is based on a simple perfect fluid.
Looking specifically in the CAA, current conditions (Figure 8a, black) indicate that the CMIP5 distribution is tri-modal: one
mode has an extent comparable to observations (at 0.6 to 0.8 of NIC extent), a second mode has a much lower extent (at 0.2 -
0.6 of NIC extent) and a third mode has an extent that covers most of the area (~1.0 of NIC extent). In the CAA, this tri-
modal distribution yields to a bi-modal distribution in 2070-2080 projections (Figure 8a, yellow): one mode still has an ex-
tent comparable to observations and a second mode has virtually no 5-month landfast ice extent left. In the Northwest Pas-
sage, the story is much simpler (Figure 8b). All considered models are entirely covered with slow, packed ice conditions at
least 5 months every year for their historical simulations but in 2070-2080 projections about half become devoid of it while
the other half retain their historical conditions. This highlights difficulty of projecting how the dynamics of sea ice will react
to anthropogenic forcing in the narrow channels of the CAA. Finally, in the Laptev Sea, almost all considered models have
little slow, packed ice extent now and by 2070-2080 (Figure 8c).
The picture is generally clearer for the CESM-LE. In that model, the CAA and the Northwest Passage has slow, packed ice
comparable to observation (Figure 8d-e). In the projection, the CAA is expected to lose only 0.2 of its slow, packed ice cov-
erage and almost none in the Northwest Passage. In the Laptev Sea, the CESM-LE is only performing marginally better that
the CMIP5 multi-model ensemble and the projection shows the complete disappearance of 5-month slow, packed ice by
2070-2080 (Figure 8f).





When we look at ice thickness, models show a wide range of ice thicknesses over landfast ice during the 1980-2015 period
for all regions (Figure 9a-c). However, for the 2070-2080 period they are essentially in agreement indicating that in all three
regions considered landfast ice thickness is found to grow between 1 and 2 meters over the cold season (Fig. 9a-c). More -
over, the projections indicate about a 0.5 m decrease in landfast ice thickness towards the end of the 21st century. A similar
growth range is apparent when just looking at the CESM-LE but there is however a larger magnitude of projected thickness
decreases towards the end of the 21st century (Figure 9d-f).

### 3.5. Ice-ocean simulations with landfast ice parameterizations

The results we have presented so far have been focused on high-resolution observational datasets, 25 km resolution reanaly -
ses and coarser climate models. From these different data sources we were able to demonstrate the capabilities and limita -
tions at emulating landfast ice conditions of models of the current generation. In the remainder of this section, we will look
at our CREG025 6 year simulations and see the benefits of using landfast ice parameterizations.

As evident in Figure 10, the CREG025 simulations show a quite accurate representation of landfast ice duration in the
Laptev Sea, the East Siberian Sea and along the Alaskan Coast where grounding is crucial for simulating landfast ice
[Lemieux et al., 2015] . The overestimation of landfast ice North of the CAA is likely a consequence of our imperfect crite -
rion for determining whether the ice is landfast or not (slow drifting ice for a NIC analyst can be identified as landfast in our
study).
Overall, in the CAA, the CREG025 landfast ice duration is in good agreement with the ones of the NIC and CIS (Figure 2a-
b). In both NIC and CIS products, the duration of landfast ice is small in tidally active regions such as the Gulf of Boothia,
Prince Regent Inlet, Lancaster Sound and Foxe Basin. In accordance with observations, the CREG025 simulation (which in -
cludes explicit tides), exhibits mobile ice in these regions throughout the winter (Figure 10b). However, CREG025 underesti-
mates the landfast ice duration in Barrow Strait and north of M'Clintock.
We are currently doing a thorough investigation of the impact of tides (and the mechanisms involved) on simulated landfast
ice. This will be the subject of a future publication. Preliminary results suggest that including tides is crucial to properly sim -
ulate landfast ice in certains regions of the CAA. We speculate that the fact that many models (e.g. GFDL-ESM2G , CESM-
LE, PIOMAS) presented in this paper, overestimate landfast ice in parts of the CAA (e.g. Gulf of Boothia and Prince Regent
Inlet) is due to the absence of tides in their simulations.
Looking at time series of 5 month landfast ice extent, the CREG025 simulation follows observations very closely in the CAA
and Laptev Sea (Figure 7a,c). In the Northwest Passage, however, the CREG025 simulation leads to too litlle landfast ice
(again due to the underestimation of landfast ice in Barrow Strait and north of M 'Clintock). This could be due to the fact that
our CREG025 simulation seems to have ice thinner (and therefore weaker) than observations (see Figure 1). Overall, how -



ever, landfast ice extent in CREG025 is much more in line with observations in all three regions than most Earth system
models (shown in Figure 8).

## 4. Discussion and conclusions

In this study, we have compared the geographical distribution of landfast ice extent and duration in ocean reanalyses and
coupled climate models and to historical ice charts. To achieve this comparison, we have used slow, packed ice in models as
a proxy for landfast ice. Using this proxy we find that some current generation models provide a reasonable representation of
landfast ice conditions (e.g. PIOMAS, CESM-LE and GFDL-ESM2G) but others still have a hard time emulating landfast
ice particularly in the CAA and even more so in the Laptev Sea. Ice-ocean simulations with a grounding scheme and a modi-
fied rheology to promote arching indicate that these parameterizations have the capability to provide better projections for
seasonal economic activities in the Arctic. This is particularly important for reducing uncertainty in Arctic shipping projec-
tions based on model simulations from the current generation of models [e.g. Melia et al., 2016]
While many models do not emulate landfast ice accurately, their biases help explain why they project dramatic ice thickness
decreases in the CAA, decreases that are not supported by long observational records [Howell et al., 2016]. Specifically, in
regions with landfast ice, models tend to have very thick ice in their historical simulations that is very sensitive to anthro-
pogenic forcing. Later in the 21st century, once multi-year ice essentially disappears from the Arctic, the thickness distribu-
tion in models becomes much more in line with the thickness expected from a simple extrapolation of springtime landfast ice
thickness records of less than ~50 cm thinning over a century from typically ~2 m springtime thickness [Howell et al., 2016].
This is also observed in the projections analyzed in this study. Indeed, in the bulk of models and ensemble members in re-
gions where landfast ice lasts more than 5 months, the end-of-winter ice thickness remains between 1-2 m until the end of
21st century.
Finally, this analysis indicates that, although the sea ice cover is projected to shrink for many months and in many regions
[Laliberte et al., 2016], landfast ice should cover most of the CAA for much of the winter well past the mid-century. This
landfast ice should reasonably be expected to grow to 1.5 m each winter, meaning that by the time the ice breaks up, haz-
ardous ice floes should remain in the region for several weeks if not months every year. The presence of these hazardous ice
floes during the months with the most economic activity will likely have negative implications, especially for shipping in the
CAA. As a consequence, in order to deal with the annual replenishing of thick sea ice in the CAA, ships will probably re-
quire reinforced hull to ward off environmental disasters as the shipping season extends earlier in the season.

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





| | Arctic | | CAA | | Northwest Passage | | Laptev Sea | |
|---|---|---|---|---|---|---|---|---|
| | 1979, 2016 | 2070, 2081 | 1979, 2016 | 2070, 2081 | 1979, 2016 | 2070, 2081 | 1979, 2016 | 2070, 2081 |
| ACCESS1.0 | 0.33 | 0.10 | 0.70 | 0.15 | 1.00 | 0.11 | 0.02 | 0.00 |
| ACCESS1.3 | 0.29 | 0.02 | 0.59 | 0.03 | 0.81 | 0.00 | 0.01 | 0.00 |
| BCC-CSM1.1(m) | 0.39 | 0.17 | 0.67 | 0.29 | 0.96 | 0.33 | 0.06 | 0.01 |
| CESM-LE | 0.52 | 0.42 | 0.91 | 0.68 | 1.00 | 0.98 | 0.10 | 0.03 |
| GFDL-CM3 | 0.52 | 0.05 | 0.96 | 0.11 | 1.00 | 0.00 | 0.18 | 0.01 |
| GFDL-ESM2G | 0.63 | 0.40 | 0.99 | 0.67 | 1.00 | 0.71 | 0.29 | 0.12 |
| GFDL-ESM2M | 0.52 | 0.34 | 0.87 | 0.65 | 1.00 | 0.97 | 0.26 | 0.11 |
| MIROC5 | 0.27 | 0.00 | 0.40 | 0.00 | 0.43 | 0.00 | 0.06 | 0.00 |
| MPI-ESM-LR | 0.29 | 0.07 | 0.44 | 0.10 | 0.59 | 0.05 | 0.02 | 0.00 |
| MPI-ESM-MR | 0.30 | 0.04 | 0.51 | 0.06 | 0.67 | 0.03 | 0.03 | 0.00 |
| MRI-CGCM3 | 1.70 | 1.51 | 1.63 | 1.62 | 1.00 | 1.00 | 1.62 | 1.47 |
| MRI-ESM1 | 1.69 | 1.41 | 1.63 | 1.61 | 1.00 | 1.00 | 1.62 | 1.36 |
| NorESM1-M | 0.57 | 0.49 | 0.93 | 0.69 | 1.00 | 1.00 | 0.01 | 0.00 |

Table 1. Fraction of NIC landfast ice extent (magenta line in Fig. 2**b**) covered by slow, packed ice with a duration of more than 5 month for different models, regions and periods.



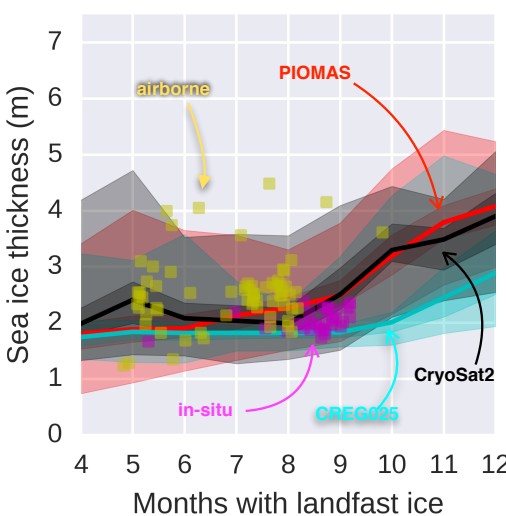

Figure 1. Canadian Arctic Archipelago (CAA) PIOMAS maximum ice thickness against landfast ice duration from Canadian Ice Service (CIS) Ice Charts over the 1980-2015 period (the mean is the thick red line, 68 one-sided percentile is the dark red shading and 95 one-sided percentile is the light red shading). In black, the same is shown for CryoSat2 instead of PIOMAS over the period 2010-2015 (see Fig. S1 for coverage). In cyan, the same is shown for the operational model CREG025 instead of PIOMAS over the years 2004-2010. In yellow scatters, the same is shown for airborne electromagnetic measurements in spring 2011 and 2015 over a small region of the CAA (see Fig. S2 for coverage). In magenta scatter, the same in shown for the in-situ CIS Ice Monitoring program at Cambridge Bay, Resolute Bay, Eureka and Alert over the period 1980-2015.



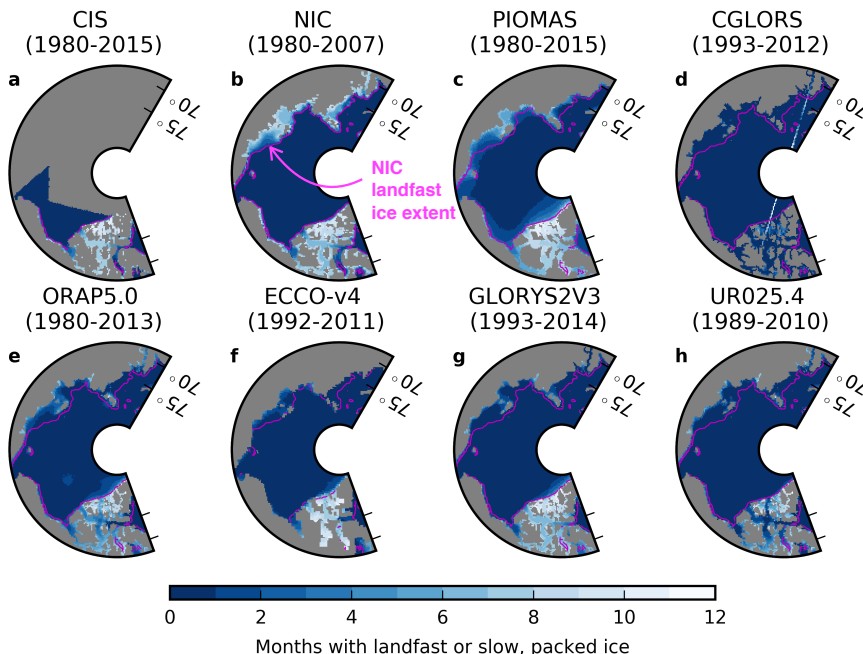

Figure 2. **a**: Historical landfast ice annual duration as reported in the CIS Ice Charts. **b**: Same as **a** but as reported in the National Ice Center (NIC) Ice Charts. **c**: Slow ( < 0.864 km day$^{-1}$), packed (> 85% concentration) ice annual duration as modeled by the assimilation product PIOMAS. **d-h**: Same as **c** but for different ocean reanalyses participating in the ORA-IP. The landfast ice extent, calculated as the 1980-2007 average one-month landfast duration contour from NIC Ice Charts, is shown in magenta.



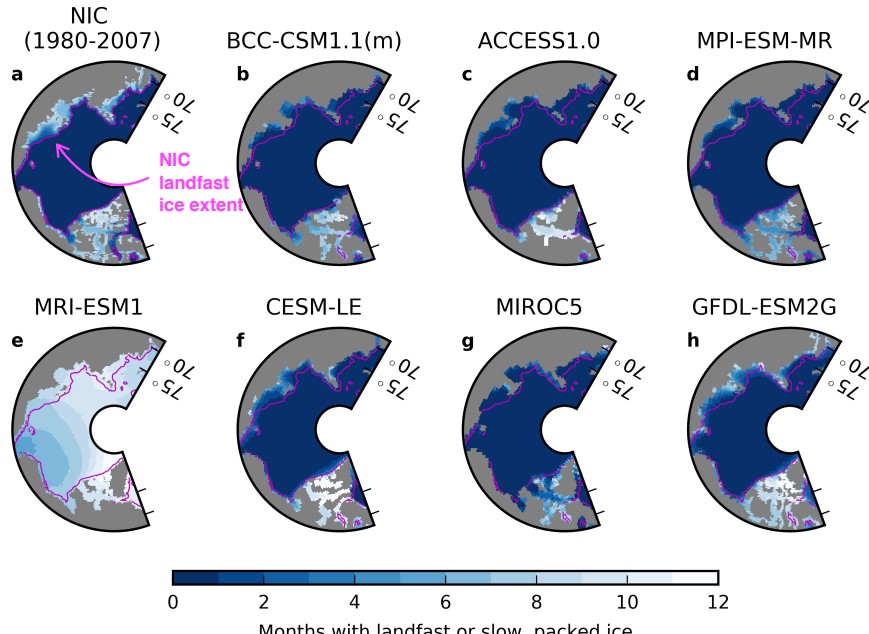

Figure 3. **a**: Same as Figure 2**b**. **b-h**: Same as Figure 2**d-h** except for a subset of simulations from the CMIP5 RCP8.5 scenario over the period 1980-2015.

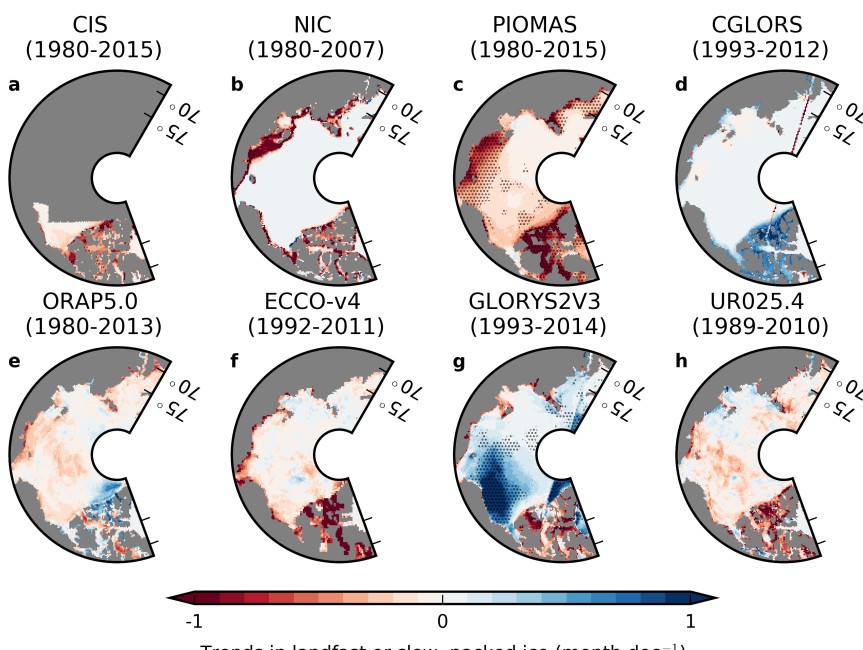

Figure 4. Same as Figure 2 but for the trends in landfast ice duration over the indicated period. Significant trends (p > 0.05) are indicated with stippling. Stippling was removed from some grid points to account for the False Discovery Rate (Wilks, 2006).

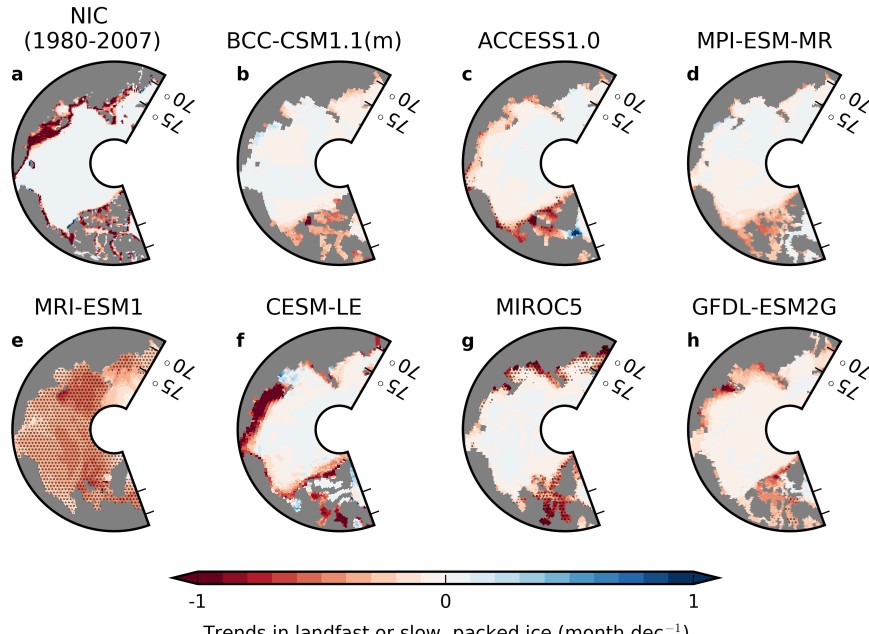

Figure 5. **a:** Same as Figure 5. **b-h:** Same as **a** but the models in Figure 3**b-h** over the period 1980-2015.





Figure 6. **a:** Time series (5 years running-mean) of the fraction of NIC landfast ice extent over the CAA (magenta line in Fig. 2**b**) covered by landfast ice from CIS ice charts for more than the number of months indicated by the line color. **b:** Same as **a** but over the Northwest Passage.

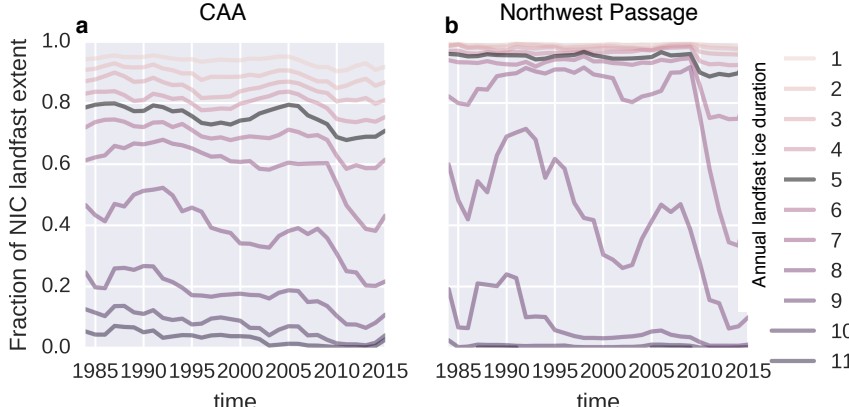

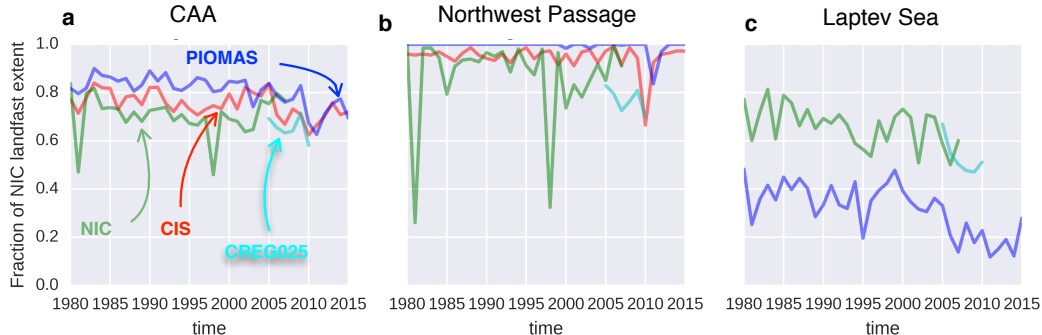

Figure 7. **a:** Time series of the fraction of NIC landfast ice extent (magenta line in Fig. 2**b**) covered by landfast ice (slow, packed ice for PIOMAS and CREG025) with a duration of more than 5 months over the CAA. **b:** Same as **a** but over the Northwest Passage. **c:** Same as **b** but over the Laptev Sea.





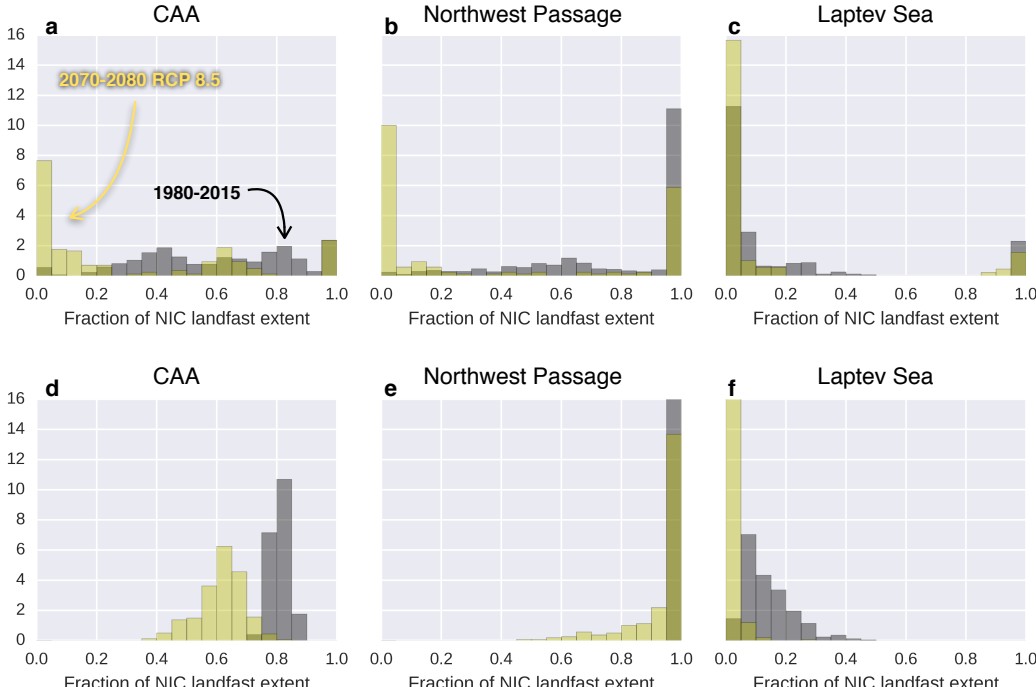

Figure 8. **a:** Distribution (across simulations and years) of the fraction of NIC landfast ice extent (magenta line in Fig. 2**b**) covered by slow, packed ice with a duration of more than 5 months over the CAA for the 1980-2015 period in black and the 2070-2080 period of the RCP 8.5 scenario in yellow. **b:** Same as **a** but over the Northwest Passage. **c:** Same as **b** but over the Laptev Sea. **d-f:** Same as **a-c** but for the CESM-LE. Note that in **e-f** the highest bins go to 21 and 19, respectively.





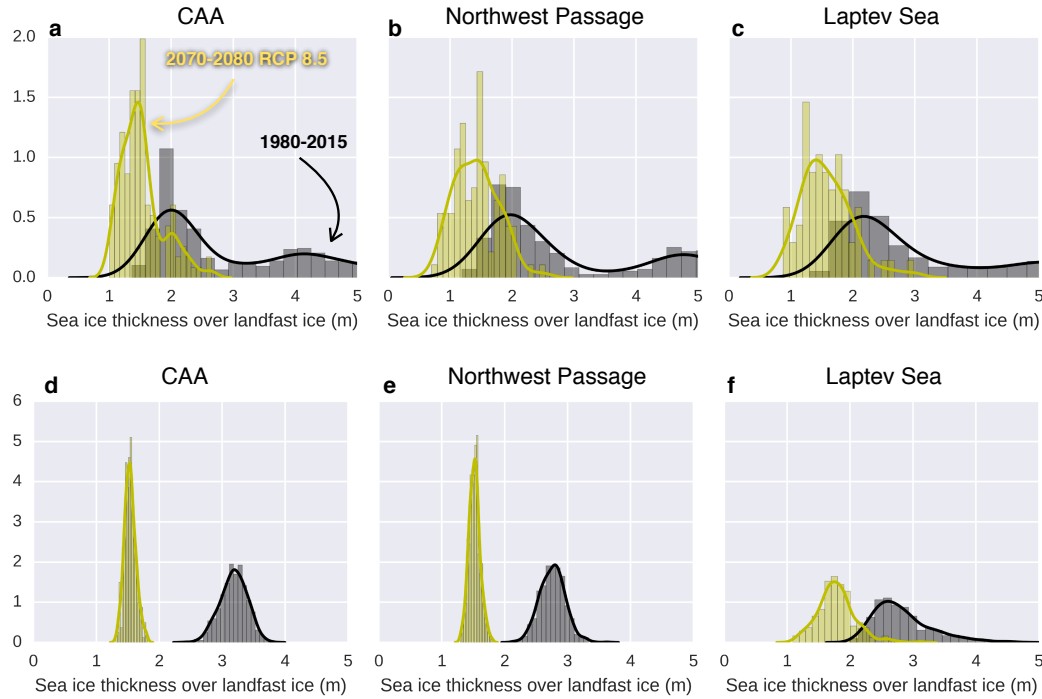

Figure 9. **a:** Distribution (across simulations and years) of the annual maximum ice thickness averaged over landfast ice duration of more than 5 months over the CAA for the 1980-2015 period in black and the 2070-2080 period of the RCP 8.5 scenario in yellow. **b:** Same as **a** but over the Northwest Passage. **c:** Same as **b** but over the Laptev Sea. **d-f:** Same as **a-c** but for the CESM-LE.



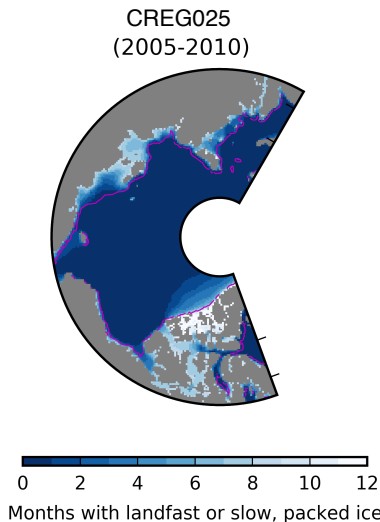

Figure 10. Same as Figure 2**b** but for the CREG025 model.