# Peer review of "What historical landfast ice observations tell us about projected ice conditions in Arctic Archipelagoes and marginal seas under anthro pogenic forcing"

_The Cryosphere, 2018_

## Referee Comment (RC1) · Anonymous Referee #1 · 17 Apr 2018

This paper examines slow and packed ice as a proxy for landfast ice in reanalysis data and climate models. The main result is that models which are closer to observations of landfast ice from 1980 to 2015 project landfast ice to remain till 2080, while models which are less realistic indicate little landfast ice by 2080. The authors conclude that winter ice conditions in the Canadian Arctic Archipelago and the Laptev Sea are very sensitive to the representation of landfast ice conditions. While interesting results are presented, there are issues regarding the methods, the rigour and the significance which needs to be addressed.

[Figure]

1. The authors derive slow and packed ice as a proxy for landfast ice in climate simulations and reanalysis data. They define "slow ice" for ice drifting less than 1 cm/s and "packed ice" for ice concentration below 85 % (weekly means). I am not convinced by the 85 % threshold. Assuming an ice thickness of 1.5 m, a snow depth of 25 cm and a surface temperature of -30 deg C, the basal ice growth rate would be around 0.5 cm/day. Over a lead with an air temperature of -30 deg C and a wind speed of 5 m/s the growth rate would be around 20 cm/day. Thus, for a lead fraction of more than 2.5 % the ice growth in the lead would dominate the basal ice growth. Thus, your slow and packed ice seems to be not a suitable proxy for landfast ice, because in contrast to landfast ice, the ice growth in the model is determined by the lead fraction and thus by sea ice dynamics. This is contradicting your bellwether argument. A higher threshold value is required and information about the sensitivity would be useful.

2. The focus of your study lies on 3 regions: the CAA, Northwest Passage and the Laptev Sea. The first two are characterized by narrow streets which cannot be resolved accurately in CMIP5 climate simulations due to limitations regarding the horizontal resolution. How meaningful is the analysis of climate models in these regions?

3. Your separation between realistic and unrealistic models and the different behaviour in climate projections is interesting, but what is the impact of the representation of landfast ice? Could it not be mainly a question how well ocean streets are resolved in the climate simulations. Some discussion about potential reasons why certain models might be more realistic would be required.

4. Large part of your conclusions is repetition of Howell et al. (2016). You refer to the study, but reference is missing under literature. The message of the added paragraph is not robust given the issues stated above.

5. The results from the ice-ocean simulation with landfast ice parameterization (3.5) are promising and extending this work (impact study by comparing simulations with and without landfast ice parameterization) could improve this paper.

---

## Referee Comment (RC2) · D. Bailey (Referee) · 11 May 2018

This manuscript describes an analysis of landfast ice (or slow compacted ice) from observations and a series of model simulations. Overall, I believe this work is important and is the right approach for comparing models and observations. That said, I have some fairly major suggestions to the authors which I believe will improve the manuscript.

1. There is always a concern about comparing apples to apples in these studies. Can

one derive a similar (slow, compacted ice) metric from the obs so this can be more directly compared to models? Also, what about the sensitivity to the thresholds of ice concentration and velocity?

2. Figure 1 is problematic for me. There is way too much going on here and the colours make it difficult to see the lines of importance. I would suggest changing the point measurments to black symbols instead of yellow and magenta. Also, the percentile ranges should be much lighter in colour.

3. Figures 2-5 and 10. Similarly, these panels are very hard to see and I would suggest just focussing on the CAA region for these. Then figure 10 could be moved into Figure 2. Plus I'm not sure you need all 8 panels. You could remove 4 panels and just talk about them in the text. This is a very qualitative picture as is, so more panels are not really necessary. Similarly with Figure 3. You could reduce this to 4 panels and then only show the CAA. Then maybe have a separate figure set of the Siberian/Laptev Seas.

4. Figure 4 is more quantitative, but could still be reduced to 4 panels and just discuss the other four in the text. I think it would be good as well for the text discussion of Figure 4 on the trends to have overall numbers for the trends in the CAA versus the Laptev. Again focussing on just the CAA and Siberian/Laptev Seas would be good here.

5. Figure 6 also cries out for some trend analysis along with significance. There is a bit of this in the text, but it could be expanded.

6. Is it possible to add the observations to the distributions in Figures 8 and 9. It would be good to know where in the model distributions the obs are.

7. There are a few minor grammatical errors. "their dynamic takes into account" for example. There are a number of acronyms and model parameters in the discussion and it would be good to spell these out and describe them more. In particular, k1 and k2 are what exactly in the Lemieux et al. model.

---

## Author Comment (AC1) · 28 Jun 2018

We are thankful for the reviewer's thought-provoking questions that have undoubtedly allowed us to better explain our intentions with this study. The reviewer's main concern, about using a threshold at 85%, is now directly addressed in the manuscript. To make a long story short, we would have liked to use a threshold of 100% to compare models to observation but, as explained in the manuscript this is apparently not how models behave. In particular, some models exhibit a reduction in ice concentration during the summer but this loss is not associated with more ice motion. As a result, this ice should

thus be considered slow and packed for the purpose of our analysis. We have thus concluded when designing this study that the better approach was to take a simple threshold that would allow our results to be reproduced while not mischaracterising model behaviours. We have chosen 85% by symmetry with the 15% used for basic uncertainty associated with low ice concentration. We have also reformulated what we meant about sea ice dynamics. In the context of our study, it was meant to include only the large-scale sea ice dynamics. Therefore, in order to make this connection explicit and in order to limit the scope of our conclusions we are now only talking about the import / export of sea ice and not sea ice dynamics in its general sense. We hope that these will clarify points 1 and 4 of the reviewer's comments.

For point 2, we believe that this paper is an attempt at addressing this issue. Are models relevant for these regions? It is our impression that it will depend on the use case. In the manuscript, we indicate that models present a bi-modal distribution in behaviour and that this might make definitive conclusions about the region tricky. In particular, it asks naturally whether one should make definitive projections about the region future economic activity given that our current modelling capability does not allow us to cleanly decide which model adequately represent sea ice import / export in the region.

The reviewer's point 3 is interesting but it is our impression that it is beyond the scope of this study. Our educated guess is that the ocean base state is likely a key player in setting sea ice behaviour in the region but this guess would require a whole different approach to validate.

For point 5, the co-authors at the CMC have looked at the different parametrizations extensively during the development of their model. Some of these results, based on the McGill model, have already been published in Lemieux et al. (2015). More recent results based on the current model are available but we have decided not to include these in this study, in order to keep the focus of this study on model representation and future projections. This detailed analysis of parametrization schemes has however

already been completed and has been submitted as another publication.

---

## Author Comment (AC2) · 28 Jun 2018

We thank very much David Bailey for his helpful review. We have tried to implement as much of his concerns as possible while keeping the thread of the manuscript as close to original as possible.

His main concerns: 1. We were worried about this as well and this is why we've used as many data sources as possible. The only thing we have not included is an analysis of sea ice velocities using ice floes tracking from satellites images. While this could

have been done, it is our impression that it would have been out-of-scope for this study as these tracking analyses are not yet available over the time and spatial scale we sought. This will probably have to be the focus of a follow-up study. 2. We have fixed the figure and kept only one percentile range. We agree with the reviewer that the figure looks much better now. 3. We have removed some panels from these figures to make them easier to read. We have decided not to further split the figures between CAA / Laptev as we already had limited our sector to this region as much as possible. Since the goal of this study was to present the CAA slow, packed ice in the Arctic context it is our impression that keeping these sectors as one is probably easier for the reader. 4. See point 3 5. We have decided against following the reviewer's advice here and we have not included a trend analysis. The main reason is because the message of this figure is that the trend will depend on the number of months of slow, packed ice and is therefore probably a quite sensitive measure of change. In particular, we note a change in trend for slow, packed ice durations around 5 months, a duration for which the trend is essentially 0. Finally, figure 6b shows a rapid change of behaviour in the last decade over the Northwest Passage which could never be captured with any meaningful measure of trend. It is our impression that in such a situation it is probably better to have the reader cry out for a trend analysis (like the reviewer did) instead of providing a trend analysis that has little statistical significance. 6. We have fixed this grammatical error and we have expanded on the Lemieux et al. parameters.

---

## Author Response (AR2)

We are thankful for the reviewer's thought-provoking questions that have undoubtedly allowed us to better explain our intentions with this study. The reviewer's main concern, about using a threshold at 85%, is now directly addressed in the manuscript. To make a long story short, we would have liked to use a threshold of 100% to compare models to observation but, as explained in the manuscript this is apparently not how models behave. In particular, some models exhibit a reduction in ice concentration during the summer but this loss is not associated with more ice motion. As a result, this ice should thus be considered slow and packed for the purpose of our analysis. We have thus concluded when designing this study that the better approach was to take a simple threshold that would allow our results to be reproduced while not mischaracterizing model behaviours. We have chosen 85% by symmetry with the 15% used for basic uncertainty associated with low ice concentration. We have also reformulated what we meant about sea ice dynamics. In the context of our study, it was meant to include only the large-scale sea ice dynamics. Therefore, in order to make this connection explicit and in order to limit the scope of our conclusions we are now only talking about the import / export of sea ice and not sea ice dynamics in its general sense. We hope that these will clarify points 1 and 4 of the reviewer's comments.

Editor:
Given the 85% ice concentration limit used in the definition of landfast ice, the lead fraction could vary considerably over time within the same region defined as landfast ice. Since this lead fraction is controlled by ice dynamics, the argument that landfast ice only evolves thermodynamically (and so is a bell weather of climate change) is somewhat weakened. I think a caveat stating this would improve the paper.

Laliberté et al.
Using 100% ice concentration is simply not representative of landfast conditions in model simulations. This was the first problem we encountered when undertaking this analysis and the 85% was chosen with care. How models should represent landfast conditions compared to how they actually represent landfast conditions is a fundamental challenge. Lead fraction in the flaw leads maybe influenced by dynamics but at 15% this is likely small and elsewhere and in particularly within the Archipelago thermodynamics is more influential.  However, we agree with the Editor that adding a caveat statement in the methodology should alleviate the continued threshold concerns of Reviewer #1.

"We acknowledge that by using an 85% ice concentration to define packed ice, the lead fraction could be large at the boundary of the slow, packed ice, due to the proximity of mobile ice. In these regions, the argument presented above might break down. In this work, we will primarily focus on archipelagoes and marginals seas where this is not an issue. It is however important to keep in mind that for applications that focus on those boundary regions, this criterion might be too lenient."

Reviewer #1:
In my first review I raised several issues, in particular the method how landfast ice has been derived in climate models and the similarities with the published paper Howell et al. (2016). The authors responded to the first point by adding a paragraph explaining why they had to choose a 85%-threshold value and clarified that sea ice thickness changes are not strongly affected by ice import/export under their definition. However, this is not my main issue. As I explained in detail in my first review, lead fraction strongly controls ice growth, thus the applied method fails to identify model regions which have similar behaviour as landfast ice. The authors did not respond to or comment my second main point: "Large part of your conclusions is repetition of Howell et al. (2016) ... The message of the added paragraph is not robust given the issues stated above." (No 4 in first review). I am afraid I have to recommend to reject the paper.

Laliberté et al.
If a 100% landfast threshold is chosen the landfast extent from model simulations is not representative compared to reality. This threshold identification was the first problem we encountered when undertaking this analysis as the models to do behave such way 100% is representative and hence we went with 85%. Indeed, the concern raised by the reviewer occurs mainly in the mobile pack ice marginal zone but not within established landfast regions and especially not within Archipelagos during the winter where the back of the envelope calculations mentioned by the reviewer are not captured. We have added this caveat as suggested by the Editor that dynamics regions may not evolve by thermodynamics alone. Finally, we feel our conclusions better confirm the results of Howell et al. [2016] and are certainly not repetition.

[revised manuscript text omitted]